# Estimation of Water Quality Parameters with High-Frequency Sensors Data in a Large and Deep Reservoir

**Cunli Li [1,2], Cuiling Jiang [1,*], Guangwei Zhu [2,*], Wei Zou [2], Mengyuan Zhu [2], Hai Xu [2], Pengcheng Shi [2] and Wenyi Da [2]**

[1]  College of Hydrology and Water Resources, Hohai University, Nanjing 210098, China; 170201020001@hhu.edu.cn

[2]  Taihu Laboratory for Lake Ecosystem Research, State Key Laboratory of Lake Science and Environment, Nanjing Institute of Geography and Limnology, Chinese Academy of Sciences, Nanjing 210008, China; wzou@niglas.ac.cn (W.Z.); myzhu@niglas.ac.cn (M.Z.); hxu@niglas.ac.cn (H.X.); spc1994@126.com (P.S.); dwyzls@163.com (W.D.)

*  Correspondence: jiangcuiling@hhu.edu.cn (C.J.); gwzhu@niglas.ac.cn (G.Z.); Tel.: +86-187-5189-3558 (C.J.); +86-189-0518-5749 (G.Z.)

**Abstract:** High-frequency sensors can monitor water quality with high temporal resolution and without environmental influence. However, sensors for detecting key water quality parameters, such as total nitrogen(TN), total phosphorus(TP), and other water environmental parameters, are either not yet available or have attracted limited usage. By using a large number of high-frequency sensor and manual monitoring data, this study establishes regression equations that measure high-frequency sensor and key water quality parameters through multiple regression analysis. Results show that a high-frequency sensor can quickly and accurately estimate dynamic key water quality parameters by evaluating seven water quality parameters. An evaluation of the flux of four chemical parameters further proves that the multi-parameter sensor can efficiently estimate the key water quality parameters. However, due to the different optical properties and ecological bases of these parameters, the high-frequency sensor shows a better prediction performance for chemical parameters than for physical and biological parameters. Nevertheless, these results indicate that combining high-frequency sensor monitoring with regression equations can provide real-time and accurate water quality information that can meet the needs in water environment management and realize early warning functions.

**Keywords:** water quality; high-frequency; sensor; estimation

## 1. Introduction

Lakes and reservoirs serve as important drinking water sources and animal habitats. However, with continuous economic growth, these lakes and reservoirs face water quality problems resulting from human activities (e.g., pollution and agricultural and fishery activities) and climate change [1]. Water quality parameters are indexes that are used to evaluate water quality environments and the dynamic spatiotemporal changes in water quality. Water quality is traditionally monitored via laboratory analysis, and such monitoring allows researchers to understand and characterize different water quality parameters [2,3]. Although water quality monitoring may produce relatively accurate measurements, this process is usually time consuming and labor intensive. Therefore, the measurements are unable to provide a temporal overview of water quality. When a water quality environment is on the verge of deterioration, the long duration of obtaining water quality parameters may delay



the implementation of preventive measures to address water quality pollution. In addition, extreme weather conditions contribute to the unpredictability of water quality environment deterioration [4–6]. Despite their timeliness in obtaining meteorological information and their frequency of water sample collection, traditional monitoring programs cannot fully characterize the spatiotemporal trends in water quality concentration [7]. To obtain timely water quality parameter data and to understand the degradation process of water environments, a relatively high-frequency monitoring of water sources is required. However, this monitoring introduces challenges in water quality management.

High-frequency sensor monitoring may make up for the defects of traditional monitoring. A high-frequency sensor is a powerful tool that has been widely used in monitoring and managing water quality. This sensor not only captures spatiotemporal changes in water quality but also overcomes the limitations of traditional programs, providing results affected by the natural environment and sampling frequency [8–10]. Studies on the application of high-frequency sensors have mostly focused on fluorescent dissolved organic matter (fDOM) and colored dissolved organic matter (CDOM). fDOM and CDOM optical sensors have been initially applied in marine systems [11–13], but recent studies have shown that these sensors can also be used in freshwater environments [14–16] and as proxies for dissolved organic carbon (DOC) [17]. Niu [8] expanded the potential application of an in situ fluorescence sensor for estimating chemical oxygen demand (COD) and TP concentrations based on the empirical correlations between CDOM absorption coefficients and COD and TP concentrations. Other studies have examined the application of turbidity sensors in monitoring water environments. For instance, Lloyd [18] used high-frequency turbidity (Turb) sensor and water chemistry data to analyze the transmission of nutrients and sediments during heavy rain. In addition to fDOM, CDOM, and Turb sensors, other physical parameter sensors, such as dissolved oxygen (DO), water temperature (WT), pH, and electrical conductivity (SpCond), are also commonly used. However, some sensors for monitoring the crucial chemical and biological parameters of water quality assessment, such as TN, TP, nitrate nitrogen ($NO_3$-N), DOC, chlorophyll a (Chl*a*), and total suspended solids (TSS), face some technical problems, with their optical activity, concentration, and measurement methods being the main limiting factors.

Fortunately, these water quality parameters may be highly correlated with optically active or physical parameters, such as Turb, DO, fDOM, WT, and pH. Christensen [19] empirically and indirectly estimated TN and TP by using high-frequency sensors. Jones [2] used data from a high-frequency Turb sensor to establish an equation for evaluating the concentration of high-frequency TP and TSS in lakes. Similarly, Viviano [20] developed a method for evaluating the concentration of high-frequency TP in urbanized water bodies and discussed the possible application of alternative evaluation methods in other water bodies. Turb is often used as an important explanatory variable in the simulation of TP. This parameter is also considered the only explanatory variable in areas where Turb and TP are highly correlated. However, using a single explanatory variable to explain the relationship of the response variable may lead to huge errors. Moreover, accurately estimating the changes in the response variable is very difficult given that the changes in water quality in ecosystems often involve a combination of physical, chemical, and biological effects. In addition, previous studies have largely focused on rivers with high concentrations of water quality parameters, whereas only few studies have examined large and clean deep-water reservoirs. Understanding the relationship between the eutrophication parameters and other optical and physical parameters of large and clean deep-water reservoirs is a topic that warrants exploration. Establishing water quality parameter conversion relationships can also help in exploring water quality information and formulating water resource protection strategies.

To fill these research gaps, this study examines the case of the Xin'anjiang Reservoir, a large and clean deep-water reservoir in China. This study collects 28-month high-frequency sensor and manual monitoring data to check whether sensor parameters hold predictive significance for the important physical and chemical parameters of water quality. This paper specifically aims to (1) establish water quality parameter regression equations; (2) verify these equations; (3) estimate high-frequency changes in water quality parameters by using these equations; and (4) test the response mechanism of the

relationship between physical and chemical water quality parameters and sensor parameters and its application in managing water environments.

## 2. Data and Methods

### 2.1. Study Area

Xin'anjiang Reservoir (29°22′–29°50′ N, 118°36′–119°14′ E) is located in the western part of Zhejiang Province (Figure 1). As a national protected, key drinking water source for China's Yangtze River Delta Region, Xin'anjiang Reservoir serves at least ten million people [1]. The reservoir is a long, narrow reservoir that has many bays, and the greatest length and width of its bays are 150 km and 50 km, respectively. Xin'anjiang Reservoir has a water area of 580 km$^2$, an average depth of 30 m, a water volume of $178.4 \times 10^8$ m$^3$, when the storage water level is at its normal value of 108 m [21]. Xin'anjiang River is a major tributary of the Xin'anjiang Reservoir and comprises 60% of the total surface runoff, and the multi-year mean runoff of it is $63.2 \times 10^8$ m$^3$ [22]. The region is located in the zone of subtropical monsoon climate. It is warm and humid with abundant rainfall, distinct seasons, and sufficient sunshine. The recorded mean annual air temperature is 17 °C, and the mean annual precipitation is 1636.5 mm (1961–2014), most of which occurs between March and July [1].

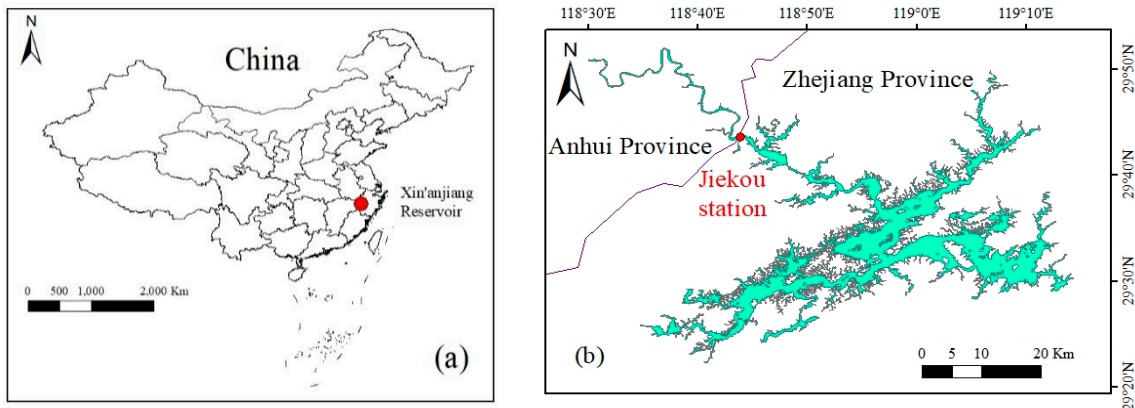

**Figure 1.** (**a**) Location of the Xin'anjiang Reservoir in China, (**b**) Location of the Jiekou station in Xin'anjiang Reservoir.

### 2.2. Data Collection

#### 2.2.1. High-Frequency Sensor Monitoring

Since April 2017, Chun'An Branch of Ecology and Environmental Bureau of Hangzhou has set YSI high frequency monitoring buoy at the Jiekou monitoring station, which monitors water quality data every 4 h. The upper water depth between 0 and 10 m is measured every 0.5 m, and the depth over 10 m is measured every 2 m. Sensor measurement parameters include water depth (WD), WT, Turb, Chla, pH, oxidation-reduction potential (ORP), phycocyanin (PC), SpCond, DO and fDOM. To ensure the accuracy of the sensors monitoring data, the probe is cleaned every two weeks.

#### 2.2.2. High-Frequency Traditional Monitoring and Laboratory Analysis

From May 2017 to July 2019, a high-frequency traditional water quality monitoring programme was carried out once every 3 days at the Jiekou monitoring station. The monitoring site was near the sensor. Water samples were taken at the upper, middle, and lower levels, with depths of 1, 3 and 5 m below the surface of the water body. Then the water was poured into the rinsed bucket, so that the sample was thoroughly mixed. Among them, 2.5 L of water sample was immediately added with 25 mL of Lugo's reagent. And after standing for 48 h, it was concentrated to 30 mL for identification

of phytoplankton community structure. The other part of the water sample was immediately frozen and brought back to the laboratory for determination of water quality parameters. The measurement parameters included TN, TP, $NO_3$-N, DOC, Chl$a$, and TSS. Among them, TN was determined by basic potassium persulfate digestion and ultraviolet (wavelength 210 nm) spectrophotometric method, TP was measured by basic potassium persulfate digestion and molybdenum antimony antichromogenic spectrophotometry (wavelength 700 nm) and the Skalar flow injection analyzer in the Netherlands automatically colorimetrically determined the water sample after GF/F membrane filtration (Skalar Co., http://www.skalar.com). The water sample was filtered by GF/F membrane (the membrane was dried and weighed in advance), and the TSS was measured by drying at 105 °C. Then, the Whatman 0.7 μm filtered fluid was analyzed in NPOC sweep mode under high temperature environment by Shimadzu TOC-L total organic carbon analyzer (Shimadzu TOC-L, Kyoto, Japan). For the measurement methods of various parameters, please refer to the following literature [23,24].

### 2.2.3. Hydrological Data

Inflow data was obtained from the Jiekou hydrological station of Xin'anjiang Reservoir, which measures inflow data once per hour (May 2017–July 2019).

### *2.3. Statistics and Analysis*

### 2.3.1. Multivariable Regression Equation

The dependent variable (response variable) is $y$, and the $j$ independent variables (explanatory variables) that affect the dependent variable are $x_1$, $x_2$, … … , $x_j$, respectively. It is assumed that the effect of each independent variable on the dependent variable $y$ is linear. Under this condition, other independent variables are unchanged, the mean value of $y$ changes uniformly with the change of independent variables, and the specific equation is:

$$y = \beta_0 + \beta_1 \times 1 + \beta_2 \times 2 + \dots + \beta_j x_j + \varepsilon \tag{1}$$

where $\beta_0, \beta_1, \dots , \beta_j$ are called regression coefficient, $\varepsilon$ is a random error term, $x$ is an explanatory variable, and $y$ is a response variable.

### 2.3.2. Estimation of Nutrient Flux

According to the characteristics of the inflow controlled by precipitation and seasons on the Xin'anjiang River, and according to the characteristics of nutrients dominated by non-point source pollution, Equation (2) was chosen for the study, which emphasizes runoff and non-point source pollution, as the calculation equation of nutrient flux in this study [25–27].

$$W = K \sum_{i=1}^{n} C_i \overline{Q}_p \tag{2}$$

where $C_i$ is the instantaneous concentration, $\overline{Q}_p$ is the average inflow during the period, and $K$ is the conversion factor for the estimated time period.

### 2.3.3. Estimation Equation Accuracy Assessment

The mean absolute percentage error (MAPE), the root mean square error (RMSE) and correlation determination coefficient ($R^2$) were used to measure and evaluate the performance of regression equations [28]. The formulas are as follows:

$$MAPE = \frac{1}{n} \sum_{i=1}^{n} \left| \frac{C_P - C_M}{C_M} \right| \times 100\% \tag{3}$$

$$RMSE = \sqrt{\frac{\sum_{i=1}^{n}(C_P - C_M)^2}{n}} \tag{4}$$

where $C_M$ is the measured concentration of the nutrient. $C_P$ is the estimated concentration of the inversion model, and $n$ is the number of samples.

2.3.4. Data Analysis

The data analysis and processing were completed by Excel 2016 (Microsoft, Redmond, WA, USA) and SPSS (Statistical Product and Service Solutions) version 24.0 (SPSS Inc., Chicago, IL, USA) and the graphic drawing was completed by Origin Pro 2020 (OriginLab Corporation, Northampton, MA, USA).

# 3. Results

## 3.1. Water Quality Parameter Concentrations and Correlation Analysis

Table 1 summarizes the variability of in situ water quality parameters during the study period. TN ranged from 0.75 mg·L$^{-1}$ to 2.33 mg·L$^{-1}$ with a mean (±standard deviation) of 1.33 ± 0.35 mg·L$^{-1}$ in the aggregated dataset. TP varied 12-fold, ranging from 11.9 µg·L$^{-1}$ to 138.78 µg·L$^{-1}$ with a mean of 11.9 ± 138.78 µg·L$^{-1}$. NO$_3$-N ranged from 0.30 mg·L$^{-1}$ to 1.76 mg·L$^{-1}$ with a mean of 0.99 ± 0.33 mg·L$^{-1}$. DOC ranged from 0.73 mg·L$^{-1}$ to 2.79 mg·L$^{-1}$ with a mean of 1.53 ± 0.40 mg·L$^{-1}$. Chl*a* varied 50-fold, ranging from 1.09 µg·L$^{-1}$ to 53.88 µg·L$^{-1}$ with a mean of 8.55 ± 8.68 µg·L$^{-1}$. TSS varied 475-fold, ranging from 0.05 to 23.76 mg·L$^{-1}$ with an average of 207.99 ± 111.31 mg·L$^{-1}$. The water quality in the Xin'anjiang Reservoir was generally better in autumn and winter than in spring and summer.

**Table 1.** Summary of the program water quality parameters in the Jiekou Monitoring Station.

| Parameters | $n$ | Mean ± Standard Deviation | Range | Season of Maximum | | Season of Minimum | |
|---|---|---|---|---|---|---|---|
| | | | | Maximum | Season | Minimum | Season |
| TN (mg·L$^{-1}$) | 256 | 1.33 ± 0.35 | 0.75–2.33 | 1.65 ± 0.25 | spring | 0.98 ± 0.16 | autumn |
| TP (µg·L$^{-1}$) | 256 | 54.45 ± 24.34 | 11.9–138.78 | 75.00 ± 17.42 | spring | 34.45 ± 18.90 | winter |
| NO$_3$-N (mg·L$^{-1}$) | 256 | 0.99 ± 0.33 | 0.30–1.76 | 1.30 ± 0.21 | spring | 0.66 ± 0.21 | autumn |
| DOC (mg·L$^{-1}$) | 256 | 1.53 ± 0.4 | 0.73–2.79 | 1.70 ± 0.43 | summer | 1.32 ± 0.33 | winter |
| Chl*a* (µg·L$^{-1}$) | 216 | 8.55 ± 8.68 | 1.09–53.88 | 11.95 ± 9.23 | spring | 133.38 ± 49.17 | winter |
| TSS (mg·L$^{-1}$) | 160 | 4.54 ± 4.11 | 0.05–23.76 | 6.16 ± 4.15 | summer | 2.58 ± 1.37 | autumn |

Table 2 shows positive correlations between TN and fDOM (0.50, $p < 0.01$), between TP and fDOM (0.49, $p < 0.01$), and between NO$_3$-N and fDOM (0.47, $p < 0.01$). Given that NO$_3$-N and PO$_4$-P are weakly acidic, their pH decreases along with an increasing concentration. Therefore, TN ($-0.38$, $p < 0.01$) and NO$_3$-N ($-0.62$, $p < 0.01$) are negatively correlated with pH. Meanwhile, DOC (0.52, $p < 0.01$) and Chl*a* (0.44, $p < 0.01$) are positively correlated with WT and are weakly correlated with SpCond. A higher precipitation increases both TSS and Turb, so a strong relationship can be observed between TSS and Turb (0.84, $p < 0.01$).

**Table 2.** Pearson correlation coefficients between water quality parameters and high-frequency sensor parameters.

| Parameters | WT | pH | ORP | SpCond | DO | Turb | Chl*a* | PC | fDOM |
|---|---|---|---|---|---|---|---|---|---|
| TN | −0.26 ** | −0.38 ** | 0.13 * | −0.36 ** | 0.21 ** | 0.32 ** | 0.26 ** | −0.34 ** | 0.50 ** |
| TP | 0.06 | −0.21 ** | 0.00 | −0.54 ** | 0.17 ** | 0.22 ** | 0.29 ** | −0.11 | 0.49 ** |
| NO$_3$-N | −0.64 ** | −0.62 ** | 0.44 ** | −0.24 ** | 0.22 ** | 0.18 ** | 0.28 ** | −0.26 ** | 0.47 ** |
| DOC | 0.52 ** | 0.33 ** | −0.43 ** | 0.11 | −0.22 ** | 0.13 * | −0.29 ** | −0.09 | −0.03 |
| Chl*a* | 0.44 ** | 0.45 ** | −0.56 ** | −0.28 ** | 0.37 ** | −0.08 | 0.46 ** | 0.10 | −0.13 |
| TSS | 0.00 | −0.12 | 0.03 | −0.49 ** | 0.17 * | 0.84 ** | 0.18 * | 0.15 | 0.36 ** |

Notes: ** $p < 0.01$; * $p < 0.05$ (two-tailed test).

### 3.2. Establishment and Verification of Regression Equations

To guarantee the reliability and practicality of the regression equations, two-thirds of the data were extracted according to the monitoring time sequence, and regression equations were established by using multiple stepwise regression methods. These equations are reported in Table 3. Table 3 also gives the *p*-values for all the regression equations and the coefficient of determination for the regressions. The larger the coefficient of determination, the stronger the correlation between the explanatory variable and the dependent variable. It should be noted that if the value of the coefficient of determination is 0, it indicates that the explanatory variable and the dependent variable are not linearly related, however, it may be related in other ways.

**Table 3.** Water quality parameter regression equations established by stepwise regression analysis method.

| Parameters | Linear Regression Analysis Results | Coefficient of Determination ($R^2$) | *n* | *p* |
|:---:|:---:|:---:|:---:|:---:|
| TN | TN = 1.088 + 0.056 × fDOM + 0.007 × Turb − 0.011 × WT − 0.113 × PC − 0.115 × SpCond + 0.051 × DO | 0.64 | 167 | $p < 0.001$ |
| TP | TP = 52.801 − 13.462 × SpCond + 3.633 × fDOM − 3.727 × PC + 0.47 × Chl*a* + 0.229 × Turb | 0.50 | 167 | $p < 0.05$ |
| NO$_3$-N | NO$_3$-N = 1.287 − 0.031 × WT + 0.052 × fDOM + 0.005 × Chl*a* + 0.002 × Turb − 0.095 × SpCond − 0.068 × PC + 0.02 × DO | 0.79 | 167 | $p < 0.05$ |
| DOC | DOC = 2.41 + 0.028 × WT − 0.01 × Chl*a* − 2.166 × ORP + 0.004 × Turb + 0.129 × SpCond − 0.122 × pH | 0.45 | 167 | $p < 0.001$ |
| Chl*a* | Chl*a* = 10.92 − 47.311 × ORP + 0.477 × Chl*a* + 0.451 × WT − 1.175 × PC | 0.56 | 140 | $p < 0.05$ |
| TSS | TSS = 3.827 + 0.708 × Turb − 7.654 × ORP | 0.68 | 100 | $p < 0.05$ |

To verify the performance of these equations, the remaining one-third of the sensor parameters were plugged into each regression equation to simulate seven water quality parameters. The results were compared with the measured values afterward. Figure 2 presents a scatter diagram of the estimated and measured values from the equation. To further verify their performance, the established regression equations were used to estimate the flux of four chemical water quality parameters. The average water quality parameter concentration within 5 m represents the concentration of an entire water column. During the experiment, the product of the water quality concentration value measured via traditional monitoring was checked every three days, and the inflow measured in this duration was treated as the measured nutrient flux. Meanwhile, the water concentration value calculated by using the regression equations and the three-day inflow value was treated as the estimated nutrient flux. Equation (4) computes for the inflow nutrient flux of the water quality parameters in the Jiekou Monitoring Station. Similar to inflow, the nutrient flux of TN, TP, NO$_3$-N, and DOC shows seasonal trends (Figure S1). Figure 3 presents a scatter diagram of the measured and simulated fluxes.

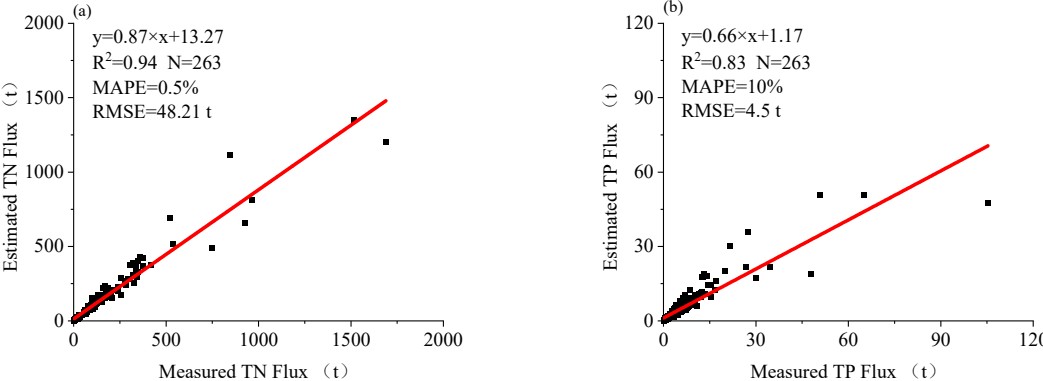

**Figure 2.** Scatter plots show the water quality parameters between the measurements and estimated values by regression equations. (**a**) TN; (**b**) TP; (**c**) NO$_3$-N; (**d**) DOC; (**e**) Chl*a*; (**f**) TSS.

**Figure 3.** *Cont.*

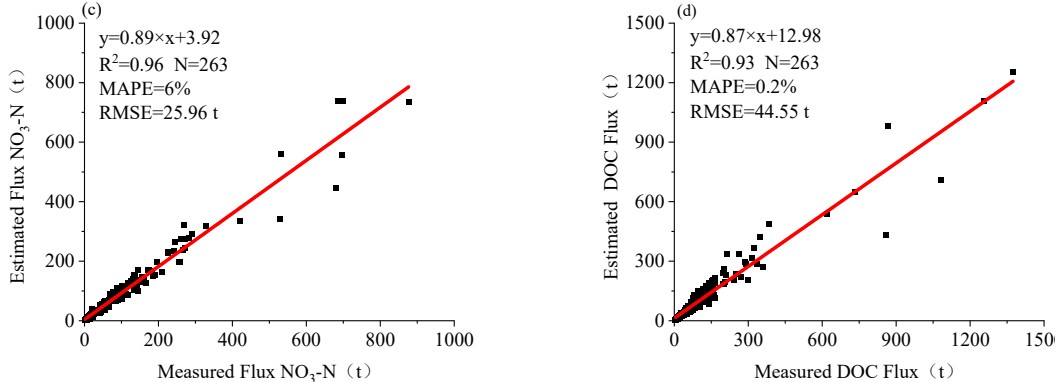

**Figure 3.** Scatter plots show the nutrient flux between the measurements and estimated values by regression equations. (**a**) TN; (**b**) TP; (**c**) NO$_3$-N; (**d**) DOC.

### 3.3. Results of the Regression Equations

The regression equations can accurately estimate the changes in various water quality parameters, but some errors are observed in certain periods. MAPE indicates the difference between the estimated and measured values (Figure 4). Interestingly, the estimated chemical parameters are significantly better than the biological and physical parameters.

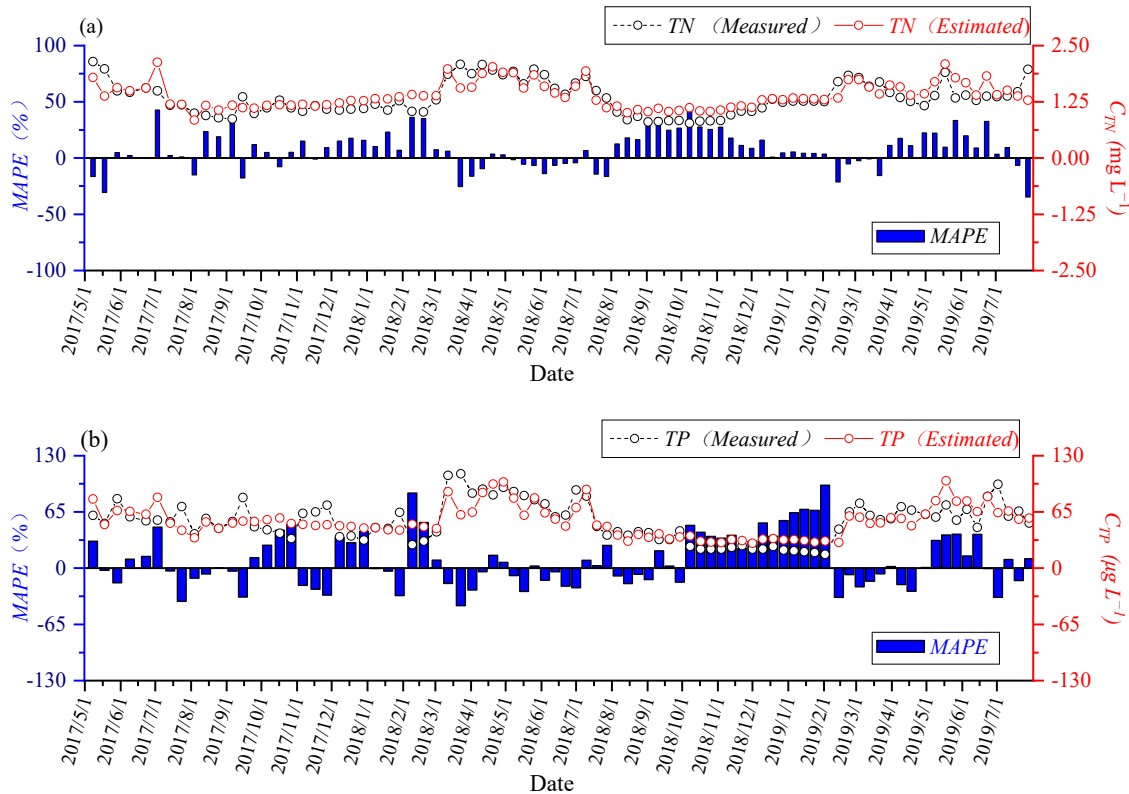

**Figure 4.** *Cont.*

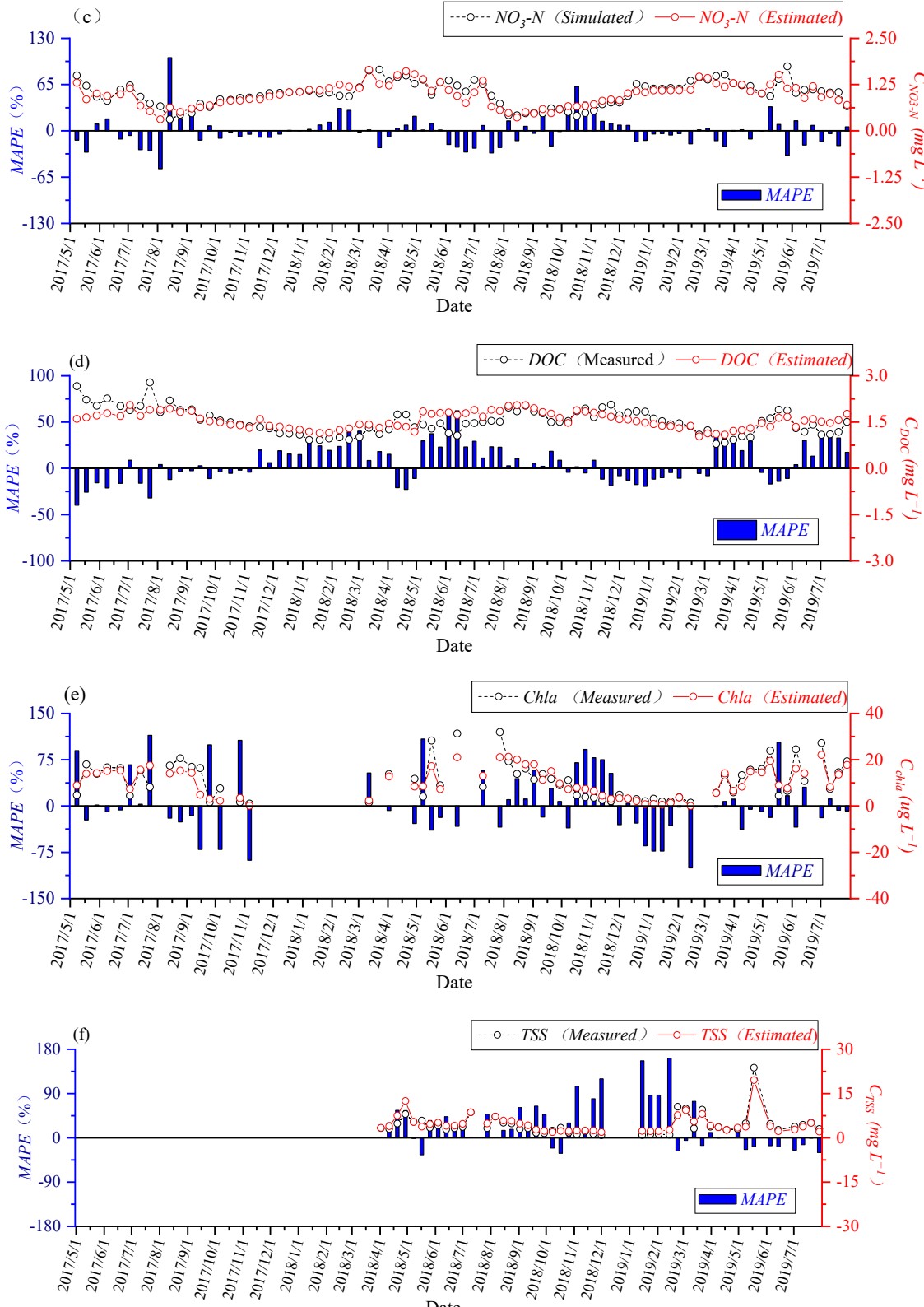

**Figure 4.** Temporal variations of estimated concentration, measured concentration and MAPE. (**a**) TN; (**b**) TP; (**c**) NO$_3$-N; (**d**) DOC; (**e**) Chl*a*; (**f**) TSS.

## 4. Discussion

### 4.1. Explanatory Variables and Influencing Factors

Sensors estimate the physicochemical parameters in water bodies based on the correlation between water quality parameters and optically active substances and the ecological correlation of water quality with sensor parameters [29]. Therefore, the correlation between water quality physicochemical parameters and sensor parameters is analyzed (Table 2). Those variables with an absolute value of correlation coefficient of >0.50 are considered strongly correlated, whereas those variables with an absolute value of correlation coefficient ranging from 0.35 to 0.50 are considered moderately correlated [30]. Consistent with the results shown in Table 2, those sensor parameters with high absolute value of correlation coefficients have great explanatory power for each response variable (Table 3). However, not all regression equations of water quality parameters have the same explanatory variables because each parameter has unique optical and ecological characteristics.

TN, TP, and $NO_3$-N all show strong correlations with the optically active substance fDOM [8]. Due to the dilution effect, SpCond is inversely proportional to TN, TP, and $NO_3$-N at a high flow [19]. TN and TP contain PTN and PTP, which are mainly attached to suspended matter. Some nitrogen and phosphorus mainly exist in particulate form, thereby suggesting that PTN and PTP adhere to particulates in water [3,31]. Therefore, fDOM and Turb are considered important explanatory variables in the regression equations. The source of DOC is mainly the primary productivity of plants, which is higher in seasons with a high temperature; in this case, WT is considered the most important explanatory variable for DOC [32]. However, the relationship between DOC and fDOM differs across the literature [29,33]. Given the positive correlation between TSS and Turb ($r = 0.84$, $p < 0.01$), the latter acts as the explanatory variable for the physical parameters [34,35]. Moreover, the Chl*a* measured via the traditional monitoring program is strongly correlated with the Chl*a*, WT, and PC measured by sensors. Therefore, these three parameters are considered important explanatory variables for the changing trend of Chl*a*.

Using multiple sensor parameters to estimate a single water quality parameter at the same time can avoid the estimation errors resulting from the abnormal value of a single sensor parameter. The estimation results of these parameters can also be used to accurately estimate the polluted substances. Given that the Xin'anjiang Reservoir is a large and clean deep-water reservoir, the water quality parameters examined in this study do not show strong correlations with a certain sensor parameter (Table 2). The correlation coefficient between Turb and TSS is the strongest ($r = 0.84$, $p < 0.01$), and the correlation coefficient between Turb and TP is far weaker ($r = 0.22$, $p < 0.05$) than those between Turb and TSS ($r = 0.95$) and between Turb and TP ($r = 0.95$) in the Little Bear Basin in Northern Utah, USA [2]. Viviano [20] compared the relationships between Turb and TP in metropolitan and urban watersheds and found that TP cannot be interpreted by Turb alone in metropolitan watersheds. This view suggests that using only one sensor parameter to estimate a certain water quality parameter is difficult. As shown in Table 3, some sensor parameters that are weakly correlated with water quality parameters are selected as explanatory variables for the regression equations in the form of regulatory variables, thereby increasing the fitting degree of these equations and improving their estimation results.

To intuitively express the accuracy of the regression equations, the evaluated MAPE is divided into five levels (Figure 5). The chemical parameters show a better estimated effect than the physical and biological parameters because the characteristics of various water quality parameters affect the sensitivity of the instruments. TN, $NO_3$-N, TP, and DOC are mostly in a soluble state in water bodies, and their distribution is relatively uniform. In most cases, TSS and Chl*a*, which are distributed in the form of particles and aggregates with an uneven size and density in water, cannot be evenly distributed at the cross-section of the reservoir [3]. This phenomenon affects not only the instantaneous data obtained by water quality sensors but also the bias resulting from manual sampling, especially in large rivers such as the Xin'anjiang River [36]. Huge differences can also be observed in the fitting effects of chemical water quality parameters, which can prove the study point of view. For instance,

the estimation effects of $NO_3$-N, TN, and DOC are better than those of TP because PTP accounts for a higher proportion of TP than $NO_3$-N, TN, and DOC [31].

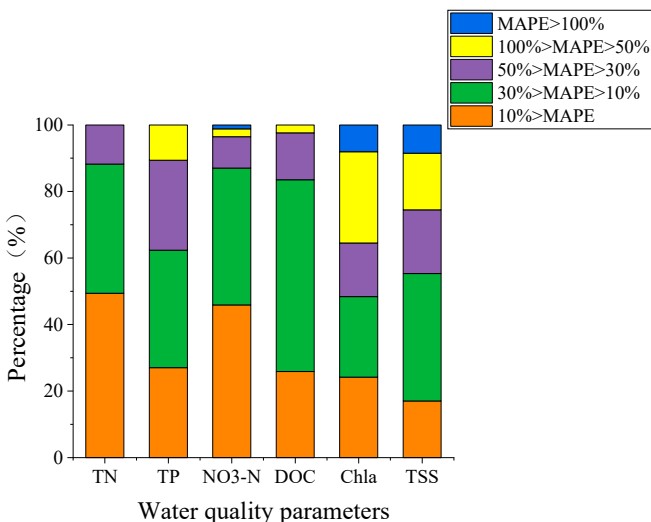

**Figure 5.** Stacked plots of different MAPE percentages of water quality parameters.

### 4.2. Evaluation of the Estimation Results

Figure 4 shows that MAPE mainly occurs when the water quality parameters show obvious fluctuations and when they reach their lowest point in a year. The sensitivity of sensors and the attributes of water quality parameters can shed light on why MAPE occurs when the water quality parameters fluctuate [37]. In seasons when these parameters easily fluctuate, sensors cannot easily capture the changes in these parameters in time. Meanwhile, the shortcomings of the regression equations can explain why MAPE occurs when these water quality parameters reach their lowest values in a year. The regression equations established via multivariate stepwise regression analysis tend to overestimate and underestimate small and large values, respectively [38,39]. For instance, the lowest mean TN values are recorded during autumn (0.98 ± 0.16 mg/L), and their MAPE value during this season is greater than those recorded in other seasons (Figure 4a).

Previous studies have employed some alternative methods to estimate the concentration of water quality parameters. One of such methods is the artificial neural network method [38,40,41], which differs from traditional regression analysis methods [19,39,42]. Experimental programs also employ traditional sample collection methods and high-frequency sensors to obtain information on the water quality parameters. Given the differences in experimental environments, water quality characteristics, sources of nutrients, and perspectives of scholars in evaluating alternative models, the advantages and disadvantages of these alternative measures cannot be easily compared.

In consideration of the number of samples and *p*-value, this study evaluates the accuracy of the proposed regression equations based on MAPE, RMSE, and $R^2$ and then compares the results with those presented in the literature. Viviano [20] constructed a TP regression equation by using Turb ($p < 0.001$) in natural environments. This equation has an RMSE of 4 $\mu g \cdot L^{-1}$, $R^2$ of 0.85, and TP ranging from 3 $\mu g \cdot L^{-1}$ to 46 $\mu g \cdot L^{-1}$. Meanwhile, the proposed regression equations in this paper ($p < 0.05$) have an RMSE of 14.32 $\mu g \cdot L^{-1}$, $R^2$ of 0.61, and TP ranging from 11.9 $\mu g \cdot L^{-1}$ to 138.78 $\mu g \cdot L^{-1}$ (Table 1). The RMSE and $R^2$ of the proposed equations are slightly inferior to those reported in Lambrone watershed, because the wide range of water quality parameters considered in this work increases the difficulty of obtaining accurate estimations by using the substitution equation. In Lambrone watershed, the TP in the metropolitan watershed ranges from 32 $\mu g \cdot L^{-1}$ to 267 $\mu g \cdot L^{-1}$, whereas RMSE increases to 32 $\mu g \cdot L^{-1}$. However, the proposed regression equations demonstrate a favorable evaluation accuracy compared with the other equations proposed in the literature [2,19].

TN is another key parameter examined in water environmental management research. Christensen [19] conducted water quality monitoring in four Kansas streams with 13, 14, 17, and 18 samples, which had corresponding $R^2$ values of 0.98, 0.83, 0.92, and 0.76, respectively. Meanwhile, the TN regression equation ($p < 0.001$) proposed in this work has an $R^2$ of 0.63 and sample number of 85. From a statistical perspective, the fitting performance of the proposed TN regression equation is slightly flawed due to the relatively large sample size considered in this work. Therefore, increasing the sample size can negatively affect the evaluation results, including the results for fitting performance. Nonetheless, the estimation results for $NO_3$-N and the evaluation results for MAPE, $R^2$, and RMSE are all satisfactory. Although no direct comparison can be found in the literature, the evaluation results for TN and TP have proven the superiority of the proposed $NO_3$-N regression equation. Previous studies have mostly focused on fDOM and CDOM spectral characteristics, which serve as proxies for DOC. The $R^2$ values reported in these studies are significantly higher than those reported in this work, but their MAPE only shows slight differences. Moreover, some studies have collected their data by using fluorescence spectrum technology, which has a slow and complex operation [33,43]. Relatively few studies have used traditional approaches to establish Chl*a* substitution methods. Meanwhile, the estimated and measured Chl*a* values in this article, which are estimated by using the substitution equation ($p < 0.05$), are relatively poor. Among the 7 water quality parameters investigated in this work, MAPE obtains the lowest value of 40%. Based on the improved artificial network algorithm, a single parameter index is used to predict the change in the concentration of Chl*a* [40], which has a MAPE of 10%. This high-frequency sensor estimate provides new ideas for predicting the value of Chl*a*. Turb is often used to evaluate the content of TSS. The estimated and measured values are compared by using the regression equations with an RMSE of 1.45 mg·L$^{-1}$. These results are deemed satisfactory compared with those of [2], who investigated samples taken from the Bear River.

The evaluation accuracy for nutrient flux also verifies the reliability of the regression equations (Figure 3). Water quality and inflow are two important factors that affect nutrient flux. Inflow plays a decisive role (Figure S2), whereas water quality shows a weak effect (Figure S3). The water in the Xin'anjiang River is mainly supplied by rainfall. Under the influence of heavy rain and runoff, a large amount of N, P, and C will enter the river, thereby increasing not only the concentration of nutrients but also the nutrients flux into the reservoir [34,44]. The $R^2$ of DOC flux and concentration is only 0.04 mainly due to the small seasonal variations in DOC concentration. Table 2 shows that the highest and lowest DOC concentrations are recorded in summer (1.70 ± 0.4 mg·L$^{-1}$) and winter (1.32 ± 0.33 mg·L$^{-1}$), respectively. In addition, due to the dilution effect, the DOC concentration may be higher in months with low inflow than in months with high inflow. Therefore, inflow cannot be easily synchronized with the inflow water concentration as reflected in the fluxes of N and P. A high $R^2$ is recorded because the concentrations of N and P greatly vary along with seasons. The regression equations solve the nutrient flux error resulting from the incompletion of the amount and concentration of incoming water, because the water quality flux recorded in a certain period is a product of both water concentration and inflow [45]. The increased or decreased water flux resulting from concentration errors in the regression equation will be compensated for later.

### 4.3. Applicability, Portability, and Limitations

Since the development and popularization of robust in situ sensors for detecting key water quality parameters, regression equations that use high-frequency sensor data have been used to characterize key water quality parameters at varying time scales (e.g., from individual events to entire years). Several studies have employed high-frequency, continuous monitoring sensors to estimate high-frequency changes in TN [19,20], TP [3,20], and TSS [35] and have discussed the applicability and portability of alternative measures of water quality parameters. Applicability and portability refer to the degree to which the developed surrogate parameter relationships are applicable by using a dataset other than that used for deriving the relationships [30]. Therefore, the type of lakes and reservoirs,

the optical water quality composition, and the environmental climate similar to that recorded in the experimental site should all be considered. The ecological environment and optical components of the Xin'anjiang Reservoir are relatively stable, and its sources of nutrients are relatively clear. Therefore, the proposed water quality parameter regression equations are theoretically suitable for similar types of reservoirs and lakes. However, the explanatory variables corresponding to different response variables warrant further discussion.

Although high-frequency sensors can solve the scientific problems related to immediateness and high frequency in the acquisition of water quality information, they have minor defects in their accuracy. Therefore, water quality managers should strictly manage the applicability of alternative methods. During seasons when water quality is highly susceptible to pollution, traditional methods can be used to obtain water quality information.

## 5. Conclusions

High-frequency sensor monitoring can aid in the high-resolution temporal observation of water quality parameters in lakes and reservoirs and has introduced new processes and ways of understanding other water quality parameters in an ecological system. Despite the growing demand, there is no key water quality parameter (i.e., TN, TP) on using these new technologies in the field, thereby inhibiting the potential adoption and application of these new technologies. To this end, this study proposes conversion equations for sensor and key water quality parameters. These equations can help water environment managers estimate the concentration of chemical (TN, TP, $NO_3$-N, and DOC), biological (Chl*a*), and physical parameters (TSS). However, these equations demonstrate a better performance for chemical parameters than for biological and physical parameters. The performance of these equations in estimating water quality flux is evaluated based on MAPE, $R^2$, and RMSE, and the results further confirm the reliability of high-frequency sensors in estimating water quality parameters.

This study specifically focuses on the Xin'anjiang Reservoir, which has a relatively stable optical composition and shows clear seasonal changes in its nutrient content. Therefore, the method applied in this study can be used in other similar freshwater systems and aid in the development of related strategies for water quality management. This method helps scientific researchers collect water quality data that cannot be obtained in traditional monitoring programs, to further understand the responses of lake and reservoir water quality to extreme climatic conditions, and to predict the occurrence of ecological disasters, such as cyanobacteria blooms and water pollution. Therefore, high-frequency sensor automatic monitoring has the potential to replace traditional monitoring in the future.

**Supplementary Materials:** The following are available online at http://www.mdpi.com/2073-4441/12/9/2632/s1, Figure S1: Histogram of monthly inflow and four water quality parameter flux during the experiment, Figure S2: Scatter plots show the monthly inflow and monthly nutrient flux by measurements, Figure S3: Scatter plots show the mean monthly water quality and monthly nutrient flux by measurements.

**Author Contributions:** C.L. writing—reviewing and editing; G.Z. experimental design and funding; C.J., W.Z., H.X., P.S., W.D., and M.Z. reviewing and editing. All authors have read and agreed to the published version of the manuscript.

**Funding:** This study was jointly funded by the National Natural Science Foundation of China (grant 41830757), the Field Station Association project of Chinese Academy of Sciences (KFJ-SW-YW036), and the Key Scientific and Technical Innovation Project of Shandong Province (2018YFJH0902).

**Acknowledgments:** The authors would like to thank Zhixu Wu, Mingliang Liu, and Xiaorui Ye, for their participation in the field samples collection and experimental analysis.

**Conflicts of Interest:** The authors declare no conflict of interest.

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
