# Peer review of "Estimation of Water Quality Parameters with High-Frequency Sensors Data in a Large and Deep Reservoir"

_water, doi:10.3390/w12092632_

Round 1

Reviewer 1 Report

Unfortunately the authors did not provided a written and detailed response to my previous comment, which hinder the understanding of their interests and arguments. However, reading the new manuscript version it is clear that the authors significantly improved the manuscript and addressed some of my main criticisms.

I do feel however that some of issues remain, mainly related to the presentation and discussion of the regression equations presented in table 3. The authors present many statistics that evaluate the regression equation performance but fail to measure the significance of the regression coefficients and of the whole regression, namely through t-tests and F-Test.

Moreover, it is also not clear how the sensors and traditional measurements at different depths were organized and compared.

Paragraph 169-176 seems not related to table 1, as the referred figures do not match.

Line 183 - Please justify the sentence

Line 222-223 . what about negative correlation coefficients

A review of the text is highly recommended. A few examples of needed revisions are presented below:

  • The abstract is still verbose, repeating the same idea a number of times.
  • line 16 - what is hydration parameters
  • line 52 - how does one sensor capture the spatial variability of a parameter
  • line 152-153 - review sentence
  • line 178-179 - review sentence
  • line 220-221 - review sentenc

Author Response

Reviewer #1

Dear Sir or Madam,

Thank you for your useful suggestions on our manuscript, they have led us to improve the quality of our paper substantially. The detailed responses to your comments are listed below point by point.

GENERAL COMMENTS

Unfortunately the authors did not provided a written and detailed response to my previous comment, which hinder the understanding of their interests and arguments. However, reading the new manuscript version it is clear that the authors significantly improved the manuscript and addressed some of my main criticisms.

PARTICULAR COMMENTS

Comment #1

(1)I do feel however that some of issues remain, mainly related to the presentation and discussion of the regression equations presented in table 3. The authors present many statistics that evaluate the regression equation performance but fail to measure the significance of the regression coefficients and of the whole regression, namely through t-tests and F-Test.

Both regression coefficient and correlation are related to the number of the samples. In addition, the performance evaluation of regression equation is not limited to regression coefficient and correlation.Therefore, in the discussion section, the regression coefficient and correlation and other evaluation methods are discussed comprehensively.

Comment #2

(2) Moreover, it is also not clear how the sensors and traditional measurements at different depths were organized and compared.

In this study, our traditional sampling method is to take water samples at 1 m,3 m and 5 m underwater, fully mix the samples and then measure the required water quality parameters. Meanwhile, we obtain the mean values of buoy parameters at 1 m, 3 m and 5 m underwater. Finally, the regression equation of traditional measurement parameters and buoy parameters are established, and its applicability is discussed. According to this equation, we can estimate water quality parameters at different depths as long as buoy parameters are obtained. Therefore, estimates at other depth, we believe, needn`t be verified.

Comment #3

(3) Paragraph 169-176 seems not related to table 1, as the referred figures do not match.

Table 1 is the number of samples and the variation range values of main water quality parameters monitored manually in this study. Line 169-176 is a comprehensive introduction to Table 1.

Comment #4

(4) Line 183 - Please justify the sentence

The erosion of the land by the rainstorm will bring a large number of suspended matter, which will cause a large area of water turbidity.

Comment #5

(5) Line 222-223 . what about negative correlation coefficients

The correlation coefficient in the literature is positive, and absolute value may be considered in this study.

Comment #7

(6) The abstract is still verbose, repeating the same idea a number of times.

We revised the abstract and deleted the verbose sentences.

Comment #8

(7) line 16 - what is hydration parameters

We have replaced hydration parameters with Water environmental parameters and highlighted it in the text. Water environmental parameters refer to the physical, chemical and biological components of the water and their quantities.

Comment #9

(8) line 52 - how does one sensor capture the spatial variability of a parameter

Multi-parameter sensor data is recorded every 4 hours, with monitoring spacing of 0.5 m for depth of 0.1 m-10 m, and with monitoring spacing of 2 m for depth of more than 10 m.Through the conversion relationship established this time, we can later estimate the important water quality parameters of the whole section.

Comment #10

(9) line 152-153 - review sentence

We have made changes in the manuscript and highlighted it in the text.

Comment #11

(10) line 178-179 - review sentence

We have made changes in the manuscript and highlighted it in the text.

Comment #11

(11) line line 220-221 - review sentenc

We have made changes in the manuscript and highlighted it in the text.

Reviewer 2 Report

Dear Author and Editor,

Thanks for your kind invitation to review the manuscript entitled “Estimation of water quality parameters with high frequency sensors data in a large and deep reservoir” by Cun-li Li, Cui-ling Jiang, Guang-wei Zhu, Wei Zou Meng-yuan Zhu,Hai Xu, Peng-cheng Shi and Wen-yi Da.

This paper specifically aims to (1) establish water quality parameter regression equations; (2) verify these equations; (3) estimate high-frequency changes in water quality parameters by using these equations; and (4) test the response mechanism of the relationship between physical and chemical water quality parameters and sensor parameters and its application in managing water environments. My recommendation is a Minor Revision. My suggestions are as follows:

  1. Check the grammar in the name of the authors.
  2. It is difficult to identify out the general content a brief summary of each section within the introduction would certainly help the reader a lot.
  3. Section 2.1 should describe the watershed of the reservoir, area, physiographic characteristics, altimetry, geology, land use, meteorology, water regime, etc.
  4. Section 2.2 Information with graphics, tables and text must be provided on the data used. As a typical year and, if necessary, highlight the inter-annual regime. Also statistics of the data used.
  5. Figure 3: all the subfigures must have the same scale on the abscissa and ordinate axes.

Kind regards

Author Response

Reviewer #2

Dear Sir or Madam,

Thank you for your useful suggestions on our manuscript, they have led us to improve the quality of our paper substantially. The detailed responses to your comments are listed below point by point.

GENERAL COMMENTS

This paper specifically aims to (1) establish water quality parameter regression equations; (2) verify these equations; (3) estimate high-frequency changes in water quality parameters by using these equations; and (4) test the response mechanism of the relationship between physical and chemical water quality parameters and sensor parameters and its application in managing water environments. My recommendation is a Minor Revision. My suggestions are as follows:

PARTICULAR COMMENTS

Comment #1

1.Check the grammar in the name of the authors.

We have checked the grammar in the manuscript.

Comment #2

  1. It is difficult to identify out the general content a brief summary of each section within the introduction would certainly help the reader a lot.

We have made a brief summary of main part of the introduction and highlighted the sentences in the text to facilitate readers to get the information we want to express.

Comment #3

  1. Section 2.1 should describe the watershed of the reservoir, area, physiographic characteristics, altimetry, geology, land use, meteorology, water regime, etc.

We have added some information about the study area and highlighted it in the text.

Comment #4

  1. Section 2.2 Information with graphics, tables and text must be provided on the data used. As a typical year and, if necessary, highlight the inter-annual regime. Also statistics of the data used.

The information of manual sampling parameters has been shown in Table 1, and the correlation between the  manual sampling parameters and buoy parameters has been shown in Table 2. If necessary, we may add an attachment to provide the information of buoy parameters.

Comment #5

  1. Figure 3: all the subfigures must have the same scale on the abscissa and ordinate axes.

 I couldn't agree with you more, but subgraph b is an order of magnitude different from subgraph a,c and d, so we don't think they should have the same scale on the abscissa and ordinate axes.

Reviewer 3 Report

The paper “Estimation of water quality parameters with high frequency sensors data in a large and deep reservoir” presents a large number of high-frequency sensor for detecting key water quality parameters and manual monitoring data to establish regression equations that measure high-frequency sensor and key water quality parameters through multiple regression analysis. 28-month high-frequency sensor and manual monitoring data were collected within the investigation.

The paper is interesting and well-written. Nevertheless, before being published, a few changes are required.

  1. Regression equations shown in Figures 2 and 3 are of type y=a*x+b. The authors should explain why the equation presents the term ‘b’. Should not it be y=a*x because when the measured value is zero, zero should also be the estimated value. Same observation for the flux
  2. Figure 4: looking at the difference between the measured and estimated concentrations, the MAPE terms seem higher when those concentrations are close by. I don’t think there is any problem in the calculation but simply a ‘weird’ definition of the MAPE term that makes it behaving like that
  3. The authors refer to regression equations published in the literature. Have they tried to use their data to check the already published regression equations?

Author Response

Reviewer #3

Dear Sir or Madam,

Thank you for your useful suggestions on our manuscript, they have led us to improve the quality of our paper substantially. The detailed responses to your comments are listed below point by point.

GENERAL COMMENTS

The paper “Estimation of water quality parameters with high frequency sensors data in a large and deep reservoir” presents a large number of high-frequency sensor for detecting key water quality parameters and manual monitoring data to establish regression equations that measure high-frequency sensor and key water quality parameters through multiple regression analysis. 28-month high-frequency sensor and manual monitoring data were collected within the investigation.

The paper is interesting and well-written. Nevertheless, before being published, a few changes are required.

PARTICULAR COMMENTS

Comment #1

(1)Regression equations shown in Figures 2 and 3 are of type y=a*x+b. The authors should explain why the equation presents the term ‘b’. Should not it be y=a*x because when the measured value is zero, zero should also be the estimated value. Same observation for the flux.

 Because the measured value and estimated value will have a certain error

Comment #2

(2) Figure 4: looking at the difference between the measured and estimated concentrations, the MAPE terms seem higher when those concentrations are close by. I don’t think there is any problem in the calculation but simply a ‘weird’ definition of the MAPE term that makes it behaving like that

MAPE means the mean absolute percentage error, which is also applied in some remote sensing inversion models.

Comment #3

(3) The authors refer to regression equations published in the literature. Have they tried to use their data to check the already published regression equations?

Some studies that have just set up the equation yet haven't test the regression equation. Certainly, there are other studies that have examined the equation and discussed its portability.

Round 2

Reviewer 1 Report

Neither the revised version nor the author's answer have addressed my main issues.

The discussion of the results presented in table 1 is confusing. Some examples:

  • The symbols of C_TN and C_TP are used in the text while the table shows TN and TP.
  • C_TN range is not shown in the table
  • Concerning the expression "C_TP grew 12-fold from 11.9 μg/L to 138.78 μg/L with a mean of 11.9±138.78 μg/L": 
    • What do the authors mean by "grew"
    • Are the mean and stand. deviation correct ? The table shows 54.45±0.35 μg/L

The discussion of table 2 is also confusing, probably due to the quality of text. For example, the sentence "Table 2 shows a positive correlation from TN to TP, NO3–N, and fDOM" is hard to follow as the correlation is between two variables not from x to y.

The addressing and discussion of negative correlation coefficients is partially solved but in line 226 there is still a reference to higher correlation coefficients when it should "higher absolute values of the correlation coefficient".

Finally, and most important, the authors did not show that each of the regression coefficients of the linear regression equations shown in table 3 are significantly different from zero.

Author Response

PARTICULAR COMMENTS

Comment #1

(1) The discussion of the results presented in table 1 is confusing

We have made changes in the manuscript and highlighted it in the text.

Comment #2

(2) The discussion of table 2 is also confusing.

We have made changes in the manuscript and highlighted it in the text.

Comment #3

(3) The addressing and discussion of negative correlation coefficients is partially solved but in line 226 there is still a reference to higher correlation coefficients when it should "higher absolute values of the correlation coefficient".

. We have made changes in the manuscript and highlighted it in the text.

Comment #4

(4) Finally, and most important, the authors did not show that each of the regression coefficients of the linear regression equations shown in table 3 are significantly different from zero.

See line193-196.

This manuscript is a resubmission of an earlier submission. The following is a list of the peer review reports and author responses from that submission.

Round 1

Reviewer 1 Report

Line 12: Grammatical error: deteriorates. Use of ”only” may be avoided.

Line 19: Specify clearly (in terms of variable) what chemical, biological and physical parameters were measured? Why those parameters were measured? Were they explicitly suggested by national water framework directive (WFD) such as EU-WFD?

Line 24: Consider using “the study” instead of “the author”. Why authors opted for the stated gooness-of-fit statistics?

Line 28-29: Consider using “nutrient flux” only not “nutrient flux of nutrients”

Abstract: The lack of clear and concise writing (albeit due to poor choice of words, grammatical lapses) made the abstract rather poor. Authors failed to provide a clear context detailed why automatic sampler can be an alternative to traditional water quality monitoring. Results are not presented in numeric terms, Future perspective are so generic, as such can’t be attributed to this study. Considering all, the whole abstract need to be completely re-written.

Line 36: Grammatical error: have been served”??

Line 37-38: Poor choice of word is also evident here. Authors used “…with the development of economy..” which should be written as “economic development”. It is getting hard to read and conceive the manuscript. I would like to request a manuscript which has gone through proof-reading from native speaker. I am sorry but I can only review then.

Reviewer 2 Report

The manuscript presents an interesting case study where a set of regression models are derived to estimate water quality parameters from measurements obtained from automatic sensors. These models may be used to evaluate the quality status of water body when the time-consuming manual laboratory analysis is not possible or to obtain a quick estimate while the more accurate results from a laboratory are not completed.

The authors present several regression equations with R2 values ranging from 0.56 to 0.78 (Table 3). The results presented in this table should be discussed in more depth. It is not clear what is presented in columns N and P, probably the number of values used in the regression and the statistical significance of a test (which one?). Although the authors briefly mention the problem of collinearity, the t-stat values of each regression coefficient are not presented nor discussed.

The discussion on the quality of nutrient fluxes estimates is presented in a somewhat independent way from the nutrient’s concentration estimates when the latter is the product of the former times a discharge value. It seems there is a partial duplication of the results being presented. It also should be highlighted that the improved values of the coefficient of determination are most probably the result of an aggregation effect, as fluxes are presented on a monthly basis and concentration as an instantaneous measurement. Is that so??

Moreover, although the water quality samples were taken at different water levels the authors do not address this issue and the possible differences in the water quality in water column.

A thorough English review is needed as there are some confusing paragraphs, some of them pointing out what seem to be important arguments which I was not able to follow. A few examples in lines 223-229, lines 281-283, lines 509-515, lines 544-545 and lines 557-558.

The use of the expression “inversion model” as a synonym of the regression model is not adequate in my opinion. Although the case study has some similarity with the traditional inverse problem in science, the manuscript does not present an inverse problem case where measurements of a set of proxy variables is used to evaluate a set of unknown and unmeasurable casual factors. The expression “estimation equation” is also not common to describe a statistical model or a regression equation.

There are also some typos spread though out the manuscript and problems in the format of some tables (e.g. Table 1).